# A Combined Microscopy Study of the Microstructural Evolution of Ferritic Stainless Steel upon Deep Drawing: The Role of Alloy Composition

**Andrés Núñez** [1,2,*] **, Irene Collado** [1] **, María De la Mata** [2] **, Juan F. Almagro** [1] **and David L. Sales** [2]

1 Laboratory & Research Section, Technical Department, Acerinox Europa S.A.U., 11379 Palmones, Spain; irene.collado@acerinox.com (I.C.); juan.almagro@acerinox.com (J.F.A.)

2 Innanomat Group, Department of Materials Science Metallurgical Engineering and Inorganic Chemistry, Imeymat, Algeciras School of Engineering & Technology, Universidad de Cádiz, 11202 Algeciras, Spain; maria.delamata@uca.es (M.D.l.M.); david.sales@uca.es (D.L.S.)

* Correspondence: andres.nunez@acerinox.com or andres.nunez@uca.es

**Abstract:** Ferritic stainless steel (FSS) is widely used to manufacture deep-drawn products for corrosion resistance applications, being the alloy drawability strongly affected by its microstructural anisotropy. This study combines a variety of microscopy techniques enabling in-depth analyses of the microstructural evolution of two different FSSs correlated to their deep drawing performance. One of the steels has a good correspondence with the standard EN-1.4016 (AISI 430). The other is a modified version of the previous one with higher contents of the ferrite-stabilising elements Si and Cr, and lower contents of the austenite-stabilising elements C, N, and Mn. Electron Backscatter Diffraction results confirm that the microstructural properties and drawability of FSS in the deep drawing process are improved in the modified steel version. Scanning transmission electron microscopy under low-angle annular dark field conditions evidences that the deformation mechanism of FSS during deep drawing follows a microstructural distortion model based on the grain size gradient and shows a variation of the deformation texture depending on the alloy composition. This work demonstrates the potential of advanced microscopy techniques for optimising the processing and design of ferritic stainless steels, with slight variations in the alloy composition, for deep drawing applications.

**Keywords:** ferritic stainless steel; recrystallisation texture; deep drawing; deformation texture; EBSD; STEM

## 1. Introduction

Ferritic stainless steel (FSS) imposes an appealing alternative to the austenitic counterpart, since it is cheap, price-stable, and has good engineering properties. Indeed, FSSs, and, more specifically, EN 1.4016 (AISI 430), are widely used in a large range of applications, including those related to formability, stretching, and deep drawing [1]. Deep drawing is a forming process that involves stretching a metal sheet into a cup-shaped product by applying a punch force, and is commonly used to produce FSS products such as kitchen sinks, pots, pans, and cans. Therefore, improving the formability and mechanical performance of FSSs is a major challenge for researchers and engineers.

During the deep drawing process, the material undergoes a series of simultaneous tensile–compression stresses that vary depending on the region of the cup-shaped product [2]. There is a higher concentration of stress at the corners of the cup, inducing higher strain there [3]. The area that experiences the greatest thinning, where the material mainly undergoes tensile strain during the deep drawing, is the so-called "cup wall". As a result of the strain applied to shape the material, deep drawing induces significant changes in the texture and microstructure of FSSs, thus affecting their final properties [4].

One key factor in the formability and mechanical performance of FSSs is the crystallographic texture [5]. This refers to the preferred orientation of crystal grains in a polycrystalline material. Texture can influence various properties of FSSs such as strength, ductility, anisotropy, magnetic behaviour, and corrosion resistance [6]. The crystallography can also change during different processing steps such as cold rolling, annealing, and deep drawing [1], consequently modifying the material performance. Therefore, correlating likely changes in the alloy texture induced upon its processing becomes of paramount importance. Not only the preferred crystal orientation, but also other microstructural features, including the crystal grain size and shape, chemical homogeneity, or the presence of extended crystalline defects, to list a few, play critical roles in the mechanical behaviour of metal alloys [7].

Several studies have investigated the texture evolution of FSSs during forming processes using different techniques such as X-ray diffraction (XRD) [8–12], electron backscatter diffraction (EBSD), and scanning electron microscopy (SEM), [8–18] and only a small number of studies employed transmission electron microscopy (TEM) [13,19]. Some of these studies focused on cold-worked FSSs. For instance, Rodrigues et al. [8] found that the grain size and initial texture had a significant influence on the texture development and the anisotropy of Nb-stabilised FSS manufactured using two-step cold rolling. Others studied the texture evolution to understand the ridging process that FSSs may experience during forming, like Ma et al. [16], who found that this defect is avoided by preventing the formation of coarse bands and grains with the {001} component in 430 and 430 LR. However, most of these studies focused on the most popular standard grade of FSSs, AISI 430 [11,16], while others explored the texture evolution of modified grades with higher contents of Ti and Nb, which have improved corrosion resistance and weldability [8–10,13–15,17–19]. None of these works studied the influence of the slight modification of the typical alloy elements of AISI 430 standard composition, such as Si, Mn, Cr, C, and N. Moreover, there is a lack of comprehensive and comparative analysis of the texture evolution of different grades of FSSs after deep drawing and its relation to their formability and mechanical performance.

In a preliminary study [20], the crystallography of uni- and biaxially deformed AISI 430 steels was studied. According to the results, a slight modification in the content of the main alloy elements led to an increase in the drawability performance of AISI 430 and changed other microstructural phenomena, such as segregation.

In this work, the drawability of the standard AISI 430 (named 0A) and the modified AISI 430 (named 1C) is studied from a crystallographic point of view. The modifications of 1C consist of increasing ferrite-stabilising elements and decreasing austenite-stabilising elements. EBSD and Scanning TEM (STEM) are used to analyse the texture and microstructure of the specimens before and after deep drawing. Also, deep drawing cup tests are performed to evaluate the drawability and mechanical performance of the specimens.

## 2. Experimental Method

### 2.1. Materials

In this work, AISI 430 ferritic stainless steel was produced under conventional industrial conditions using an electric arc furnace, AOD converter, and continuous casting. The reduction in slab thickness took place during hot rolling, and the material was recrystallised at the end of this stage using an annealing treatment. After this, the final thickness of the coil was reached after the cold rolling stage and a final annealing process was performed in a non-oxidising atmosphere ($H_2/N_2$) at 850–860 °C (typical final annealing temperature for these materials). More details about the production process can be found at www.acerinox.com (accessed on 28 December 20023). Two different ferritic types were studied, with different chemical compositions and final thicknesses. These materials are identified as 0A, which has the standard AISI 430 chemical composition, and 1C, that shows some modifications in chemical composition and manufacturing with respect 0A. The chemical composition (% wt.), included in Table 1, of 0A ferritic stainless steel is Fe balance, 0.050 C, 0.035 N, 0.35 Si, 16.3 Cr, and 0.41 Mn, with a final thickness of 0.8 mm.

On the other hand, the chemical composition (% wt.) of 1C ferritic stainless steel is Fe balance, 0.025 C, 0.025 N, 0.45 Si, 16.7 Cr, and 0.31 Mn, with a final thickness of 0.4 mm; the materials are compared under their most commonly used geometry, since 0A steel is usually manufactured from 0.5 to 0.8 mm, whereas 1C is manufactured from 0.35 to 0.5 mm. The composition of 1C shows a higher content of ferrite-stabilising elements (Si, Cr) and a lower content of austenite-stabilising elements (C, N, Mn), which leads to a lower ferrite–austenite transformation at high temperatures compared to 0A.

**Table 1.** Chemical composition (% wt) of 0A and 1C samples.

| Element | 0A | 1C |
|---------|------|------|
| C | 0.050 | 0.025 |
| N | 0.035 | 0.025 |
| Si | 0.35 | 0.45 |
| Cr | 16.3 | 16.7 |
| Mn | 0.41 | 0.31 |

For this paper, 0A and 1C specimens were obtained and characterised after final annealing. Afterwards, deep drawing cup tests were carried out on these samples and new specimens were taken after deformation to evaluate the deep drawability.

### 2.2. Deep Drawing Cup Test (Limiting Drawing Ratio)

Deep drawing cup tests were performed to determine the deep-drawability and Limiting Drawing Ratio (LDR) of the studied materials. These tests were executed on a Zwick Roell BUP 600 sheet metal testing machine, using 66 and 68 cm of blank diameter and 33 cm of punch diameter. LDR is defined as the ratio of the drawn part diameter to the limit diameter of the blank [1]. The acceptable result corresponds to the diameter of the blank that has not reached fracture after the drawing process.

To analyse the crystallographic evolution of the samples after deep drawing, specimens were acquired from the wall of the cup, marked with a red arrow in Figure 1.

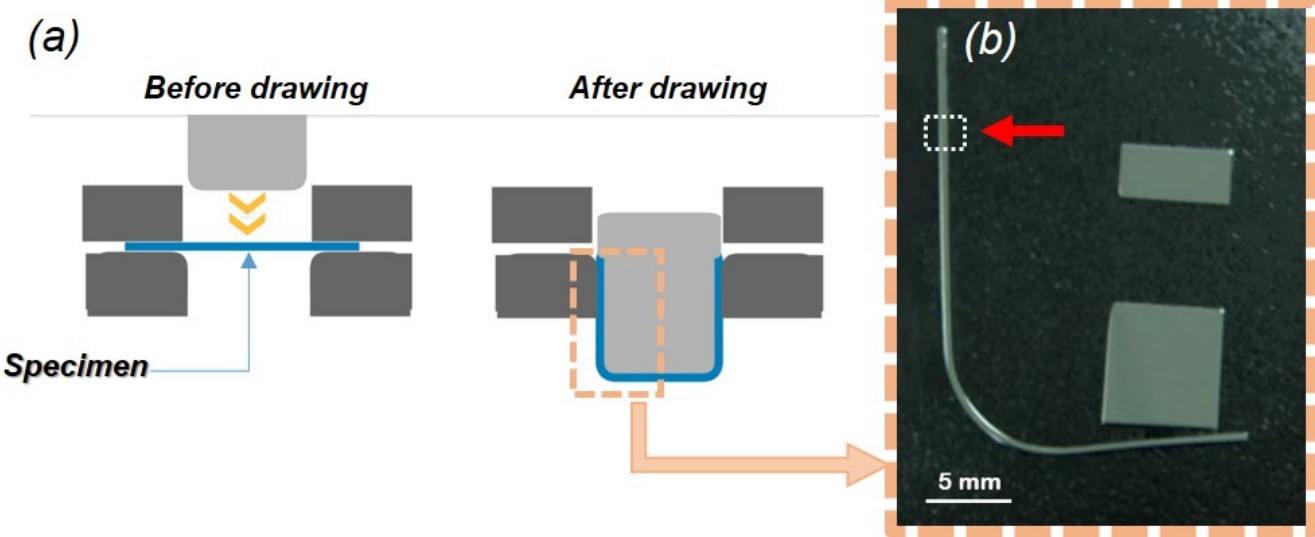

**Figure 1.** (**a**) Deep drawing cup test scheme. (**b**) Image of half of the cup after deep drawing, polished for EBSD analysis. $Y_0$ = rolling direction, $X_0$ = drawing direction.

### 2.3. Microstructure Analysis

The microstructure analysis was performed using electron backscatter diffraction (EBSD) and a scanning transmission electron microscope (STEM) on the specimens before and after the deep drawing test. The sample preparation for EBSD characterisation was carried out with a cloth disc and diamond paste by auto-polished, with a final polish of colloidal silica suspension. Rolling (annealed samples) and deformation directions (after drawing test samples) were kept for the analysis. The EBSD results were acquired using a Zeiss Ultra 55 FEG-SEM. The working conditions for the EBSD analysis were 20 kV, 16.5 mm of working distance, and 0.5 μm of step size. The post-processing of the data was carried out using the Channel 5 EBSD system (Oxford Instruments plc, Abingdon (Oxfordshire), UK) from Oxford Instruments [21].

STEM measurements were carried out to address the microstructure and chemical composition of the samples. For this analysis, lamellas were prepared by means of focused ion beam (FIB) milling using a Zeiss Crossbeam 550 FIB-SEM workstation. The lamella preparation involved a Pt deposition of 3 μm and a mill down of 15 μm. Finally, after the in situ lift-out of the lamella, the thinning procedure reached a final thickness of 150 nm. The STEM analysis was performed in a Thermo Scientist Talos F200X microscope operated at 200 kV and equipped with an XFEG gun. The equipment has two annular detectors for different imaging modes and a four-quadrant windowless energy-dispersive X-ray detector (SDD Super-X detector), achieving a solid angle of 0.8 srad. The STEM measurements were performed using a camera length of 98 mm to fulfil low-angle annular dark field (LAADF) conditions (i.e., 30–80 mrad), providing images whose contrast relates to the lattice strain and structural defects. STEM-EDX imaging was performed to obtain compositional maps at the nanometre scale.

## 3. Results and Discussion

### 3.1. Microstructure Analysis of BCC after Annealing (before Drawing Test)

The microstructure and texture components of the samples extracted after the industrial bright annealing process were evaluated. Figure 2 shows {100}, {110}, and {111} pole figures of the annealed 0A and 1C. Based on these results, the distribution of poles in the three planes of projection is typical of AISI 430 FSS [22,23]. It is observed that there is a high-density pole at <111>{111}, which is stronger in the 1C sample. Moreover, the maximum texture intensity is identified in {111}, about 4.6 and 7.4 times the random orientation distribution in the three projection planes in 0A and 1C, respectively.

On the other hand, Figure 2 depicts a symmetrical pole distribution located around the centre of the projection planes, which corresponds to texture fibre along the $Z_0$ direction. In Figure 2a, the presence of texture fibre is identified in the {110} and {111} projection planes, whereas in Figure 2b, the texture fibre is observed in the three planes. Regarding the {100} plane of Figure 2a, poles are not distributed on the circumference of this pole figure and a lack of rotational symmetry can be observed in the three projection planes. This indicates that some crystallographic distortion is present in the microstructure of the annealed 0A specimen. According to [24], the pattern pole distribution observed in the {100} plane of Figure 2a corresponds to the formation of martensite in this alloy.

On the other hand, Figure 3 displays the normal-projected ($Z_0$) IPF orientation map for the 0A and 1C samples after bright annealing. Figure 3a shows no preferred orientation of the grains in the annealed 0A sample. Further, it is observed in Figure 3b that the {111} plane includes the favoured orientations of the grains in the annealed 1C specimen. Therefore, the texture is stronger in the annealed 1C sample compared to 0A, which agrees with the results from Figure 2.

Thus, it can be determined that {111} is the rolling plane [23], parallel to the sample surface, and <111> is identified as the recrystallisation direction. Therefore, the recrystallization texture of annealed 0A and 1C can be written as {111} <111>.

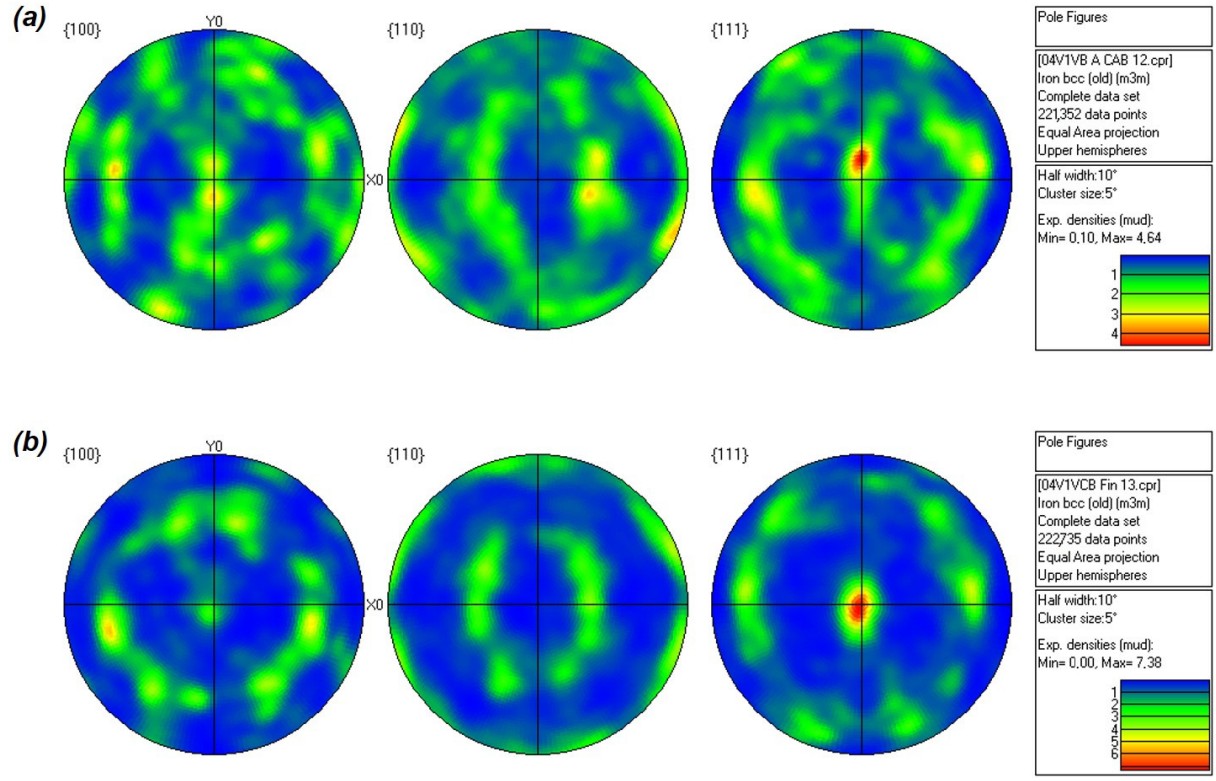

**Figure 2.** Pole figures of the annealed 0A (**a**) and 1C (**b**) samples.

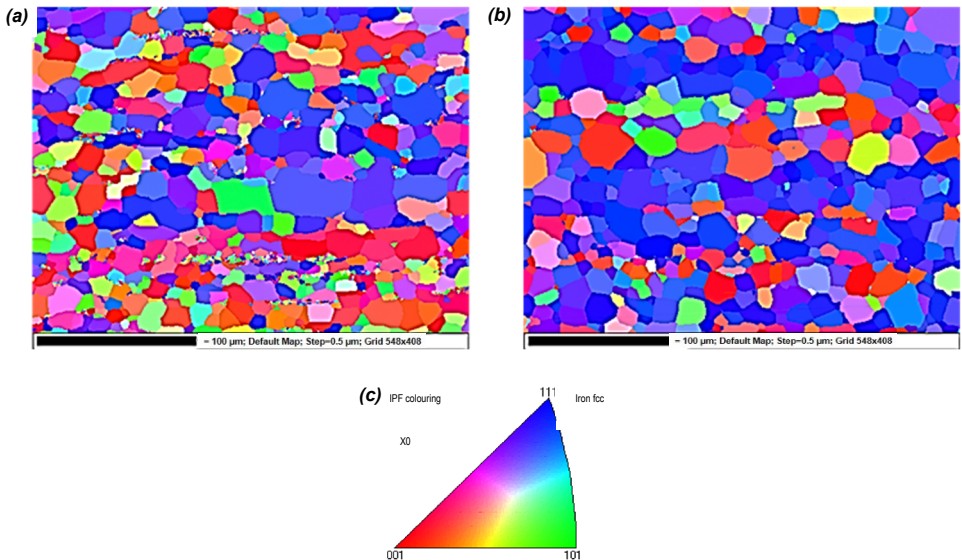

**Figure 3.** IPF orientation map from the Z-projection of the specimen surface of (**a**) 0A and (**b**) 1C, and (**c**) inverse pole figure colour map.

The texture fibre and the principal components of texture related to annealed specimens [25,26] were quantified using EBSD and Channel 5 and are listed in Table 2. The results show that the annealed specimens do not exhibit an α-fibre (also known as RD-fibre) texture [27]. However, a γ-fibre recrystallisation texture was significantly detected in both alloys. This is in good agreement with Figure 2, where it was observed that the pole distribution in the {111} plane is symmetrical around the centre of this pole or <111>. Therefore, the annealed 0A and 1C specimens mainly developed a {111} <111> γ-fibre recrystallisation texture after bright annealing [22]. The intensity of the γ-fibre recrystallisation texture was higher (more than double) in the annealed 1C than in the annealed 0A specimen.

**Table 2.** Texture fibre and components in BCC-annealed 0A and 1C samples.

|  | Annealed 0A | Annealed 1C |
|---|---|---|
| Texture fibre (%) | | |
| α-fibre | 0 | 0 |
| γ-fibre | 26.7 | 54.7 |
| η-fibre | 11.4 | 2.6 |
| <110>ND | 9.2 | 5.8 |
| <100>ND | 17.9 | 14.7 |
| <011>RD | 19.9 | 12.9 |
| Most common texture components in BCC (%) | | |
| Goss | 5.4 | 1.7 |
| Cube | 7.3 | 2.5 |

The higher presence of γ-fibre promotes the improvement of drawability in the annealed 1C sample as a result of a previous higher cold-rolling deformation bright annealing [28] (91% thickness reduction in annealed 1C versus 84% thickness reduction in annealed 0A steel). After cold deformation, the material's stored energy increases as more dislocations are created [29]. This stored energy is the driving force for γ-fibre formation during the subsequent recrystallisation [29,30] (under optimum conditions). Therefore, this suggests that the driving force was higher in annealed 1C steel due to higher cold deformation.

In addition to the orientation of the crystal grains, their sizes were measured. Figure 4 shows the size distribution of the grains in the annealed specimens based on the EBSD results. The average grain size is 8.5 μm in 0A and 10.9 μm in 1C. Figure 4a shows that the 0A specimen has many fine grains from 2 to 7 μm in diameter. In order to determine the appearance of these fine grains, a GB+CSL map was recorded from the EBSD analysis of the annealed 0A and 1C. Figure 5a shows that these fine grains correspond to regions with a high twin concentration (3-sigma, twin boundary), marked in the image with circles. Figure 5b confirms that after annealing, the formation of these areas of high twin concentration did not take place in the 1C alloy.

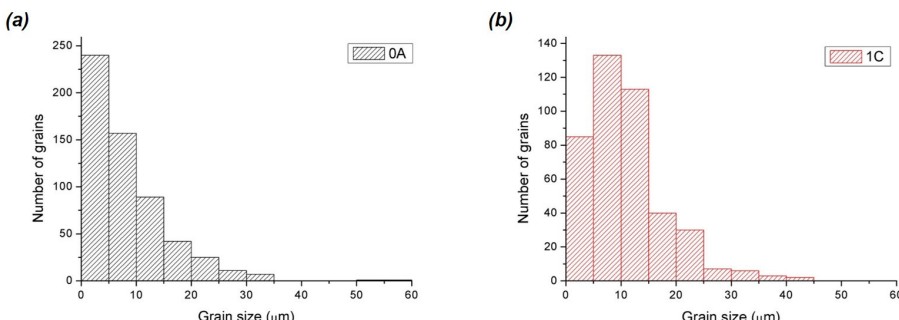

**Figure 4.** Grain size distribution in the annealed (**a**) 0A and (**b**) 1C samples.

The areas of high twin concentration were characterised at higher magnification by EBSD and LOM. In Figure 6a, the GB+CSL map shows, in detail, a high twin concentration area. According to Figure 6b, the pattern quality (band slope) map displays a clear difference in the sharpness of the Kikuchi bands inside of these regions of high twin concentration. These areas have the darkest shades of grey, which indicates smoother patterns related to higher distortion in the crystal lattice. Moreover, Figure 6c shows an LOM image of the areas of interest, which reveals the hierarchical nature of the martensitic structure, marked with arrows in the image.

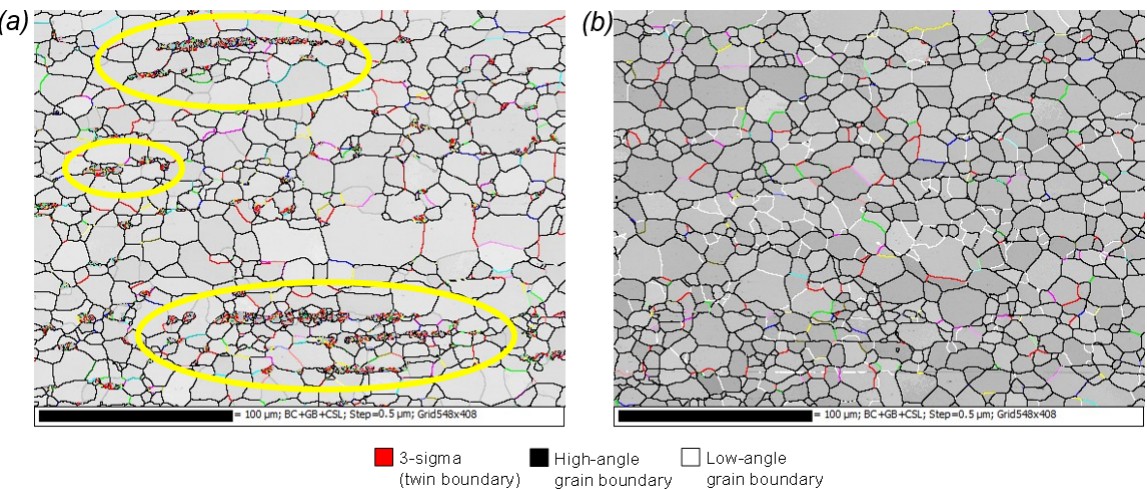

**Figure 5.** GB+CSL map of the microstructure of the annealed (**a**) 0A and (**b**) 1C specimens obtained via EBSD. 3-sigma, twin boundary, marked in the image (**a**) with yellow circles.

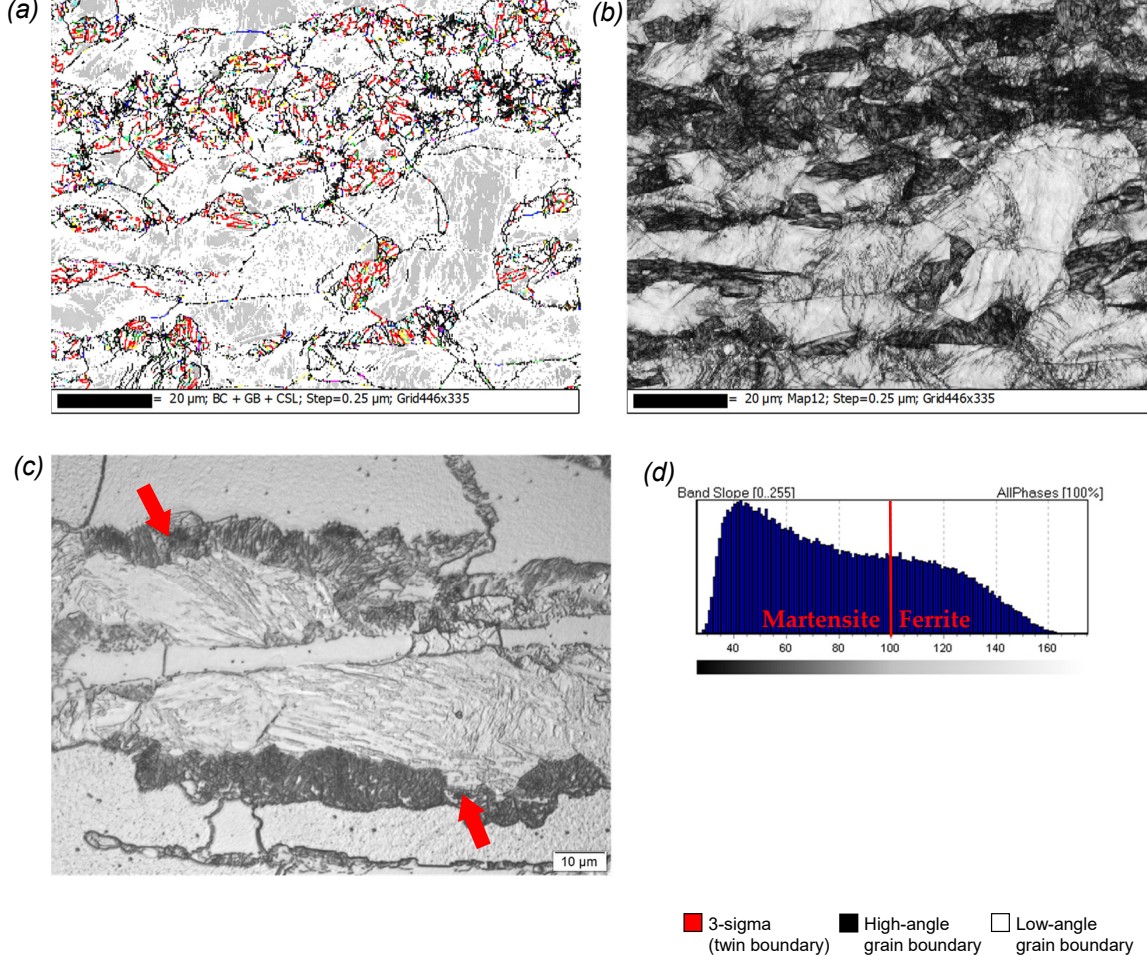

**Figure 6.** (**a**) GB+CSL map of an area with a high twin concentration of annealed 0A. (**b**) Pattern quality (band slope) map of 0A. (**c**) High-magnification LOM image of an area with high distortion of the crystal lattice of annealed 0A. (**d**) Band slope distribution function.

Furthermore, Figure 7 displays the XRD spectra of the annealed 0A and 1C alloys, which correspond to the microstructures observed in Figure 5. According to XRD, 0A and 1C showed ferritic peaks at the $(110)_\alpha$, $(200)_\alpha$, $(211)_\alpha$, and $(220)_\alpha$ diffraction positions. Specifically in the 0A sample, an austenitic peak was detected at the $(111)_\gamma$ diffraction position. In agreement with [31], this peak corresponds to the packet of the martensite morphology. On the other hand, it can be observed that there are variations in the ferritic peak intensities. The intensity of peak $(110)_\alpha$ in 0A increased after annealing compared to 1C. This variation is closely related to the martensite formation in 0A [32,33]. Overall, the results confirm that the crystallographic distortion observed in the {100} projection plane of Figure 2a is due to the presence of residual martensite in the 0A specimen. This remaining martensite in 0A after annealing resulted in a weaker γ-fibre compared to 1C, as observed in Table 2.

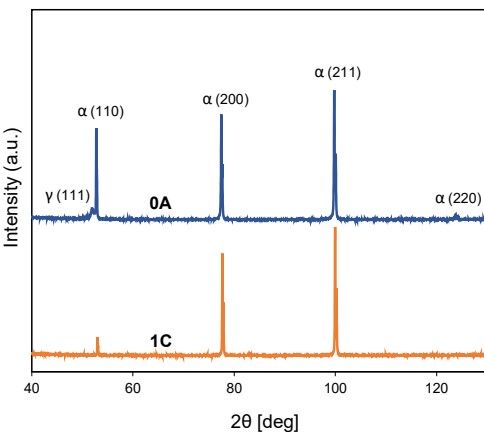

**Figure 7.** XRD spectra of annealed 0A and 1C.

## 3.2. Microstructure Analysis of BCC after Drawing Test

The results of the deep drawing cup test revealed that the LDR parameter was equal to "2" in both alloys. Considering that the 1C metal sheet was thinner than 0A, it can be assumed that the drawability is better in 1C than 0A under the same test conditions. However, in this work, the use of more advanced characterisation techniques is required to study the drawability of the analysed steels, since the information from the deep drawing cup test is insufficient.

Figure 8 displays the pole figures of deformed 0A and 1C specimens after the deep drawing cup test (sampling described in Figure 1). It can be observed that the crystallographic orientations dramatically changed after the drawing tests with respect to the crystallography displayed in Figure 2. The observed fibre texture along $Z_0$ (see Figure 2) disappeared, and a new fibre texture along $X_0$ formed after the drawing test in the 0A sample (Figure 8a). On the other hand, in Figure 8b, no fibre texture could be identified in 1C after deep drawing. However, new deformation components can be observed in the projection planes of Figure 8b (labelled as A, B, C, D, E, and F).

After deep drawing, 0A and 1C showed a common component "A" in the {110} plane, which can be written as <001>{110}. In both alloys, the intensity of this component was around 4.7 times higher than the random orientation distribution and corresponded to the Goss component. Thus, this establishes that the common deformation direction was likely <001>{110}, parallel to $X_0$. Nonetheless, in 1C, the slip system can be defined by more than one component. The angles of the components from B to F were calculated with respect to the deformation direction ($X_0$) based on the method proposed in [23]. The angles and projection components of the B, C, D, E, and F poles corresponding to the 1C drawn alloy are listed in Table 3.

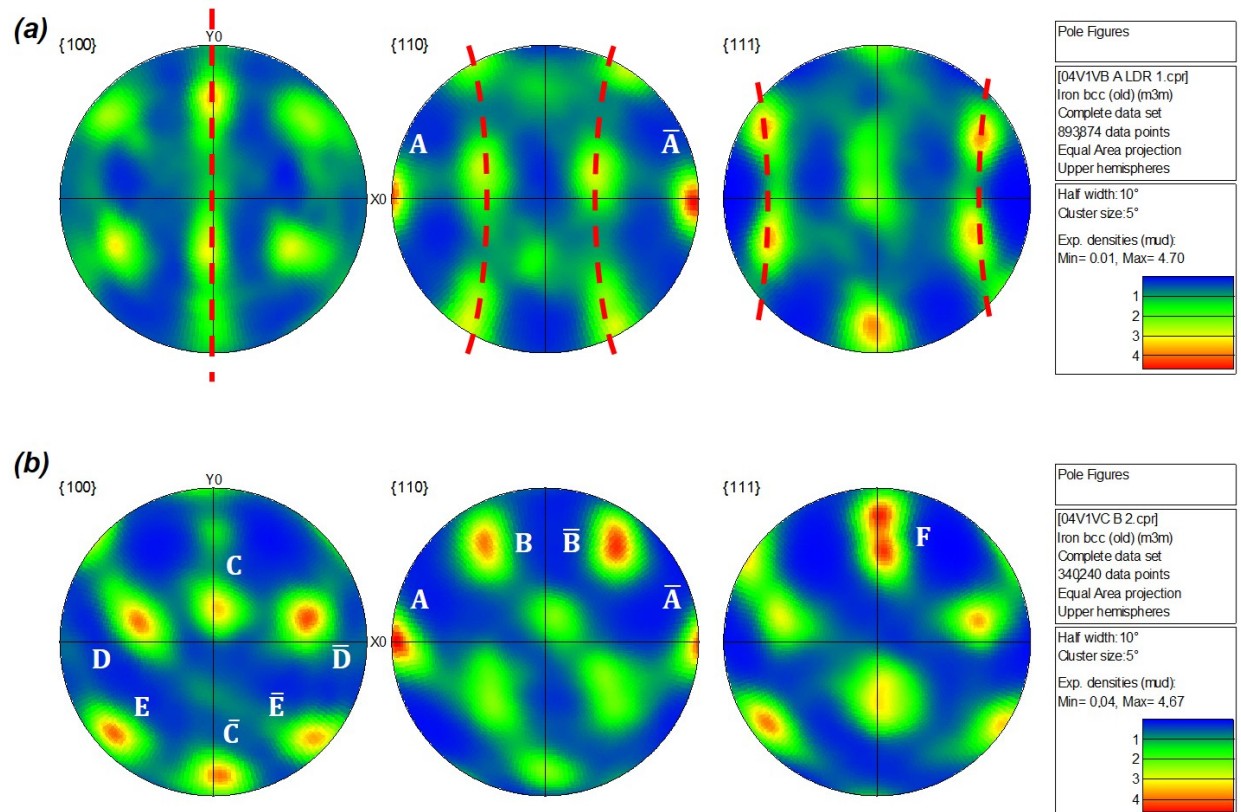

**Figure 8.** Pole figures of deformed (**a**) 0A and (**b**) 1C after drawing test. $X_0$ = drawing direction; $Y_0$ = rolling direction. Projection plan are labelled in image as A, B, C, D, E and F and its parallel $\overline{A}$, $\overline{B}$, $\overline{C}$, $\overline{D}$, and $\overline{D}$.

**Table 3.** Angle with respect to $X_0$ and projection components of B, C, D, E, and F poles in the deformed 1C specimen.

| | Drawn 1C | |
|---|---|---|
| | Angle w.r.t $X_0$ (°) | Texture Component |
| B | 73.9 | <101>{110} |
| C | 90.0 | <101>{100} |
| D | 54.6 | <111>{100} |
| E | 82.6 | <011>{100} |
| F | 90.0 | <100>{111} |

Considering the deformation direction, Figure 9 shows the $X_0$-projected IPF orientation map for the 0A and 1C samples after the drawing test. These results indicate that during the drawing test of 0A and 1C specimens, slip takes place, preferably on the {101} plane. This indicates that the {101} plane is parallel to the deformation direction <001>. Therefore, taking into account the results from Figures 8 and 9, the deformation texture of 0A can be described as <001>{110}, being the most close-packed direction of this BCC specimen [34]. In addition, in the case of the 1C alloy, the slip system can be defined as <101>{110}, <001>{110}, and <100>{111}.

Table 4 summarises the results of the texture fibre and the Goss component in the annealed and deformed 0A and 1C specimens, quantified using EBSD. It can be observed that the intensity of the γ-fibre texture considerably decreased in both specimens, while an α-fibre texture appeared after the drawing test. Moreover, the intensity of the Goss component also increased after the drawing test in both samples. Comparing both alloys after

deep drawing, it can be determined that 1C showed a higher $\alpha$-fibre intensity, along with a stronger reduction in the $\gamma$-fibre texture, and a higher intensity of the Goss component than 0A. These results evidence the better mechanical performance of the annealed 1C alloy, which indicates that the stored energy capacity is higher in 1C compared to 0A.

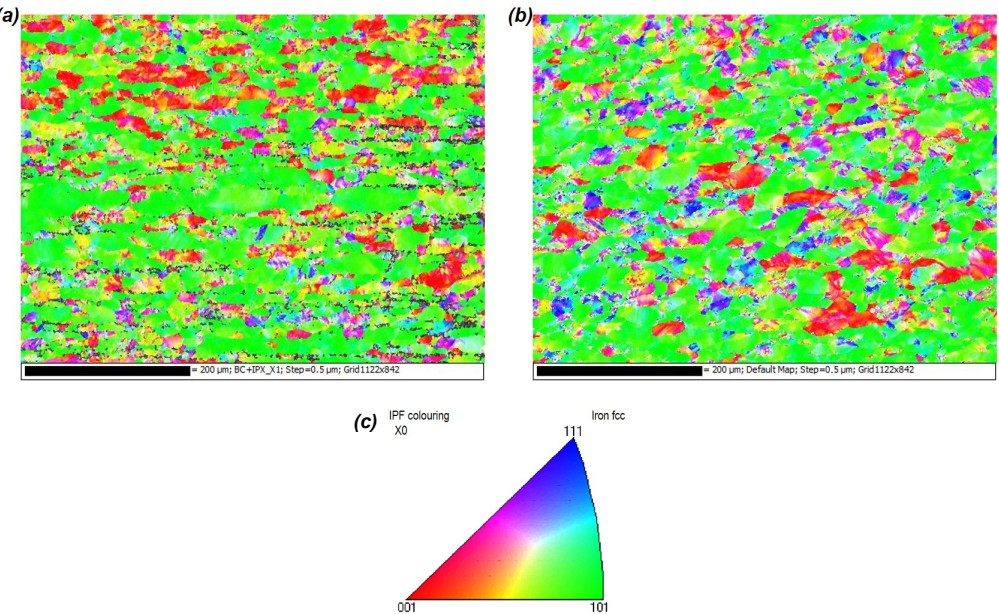

**Figure 9.** IPF orientation map from the X-projection of the specimen surface. (**a**) 0A and (**b**) 1C after drawing test.

**Table 4.** Texture fibre and Goss component in BCC-deformed 0A and 1C after deep drawing.

|  | Annealed 0A | Drawn 0A | Annealed 1C | Drawn 1C |
|---|---|---|---|---|
| **Texture fibre (%)** | | | | |
| $\alpha$-fibre | 0 | 52.9 | 0 | 65.4 |
| $\gamma$-fibre | 26.7 | 12.4 | 54.7 | 3.9 |
| $\eta$-fibre | 11.4 | 11.9 | 2.6 | 9.1 |
| <110>ND | 9.2 | 0 | 5.8 | 0 |
| <100>ND | 17.9 | 0 | 14.7 | 0 |
| <011>RD | 19.9 | 0 | 12.9 | 0 |
| **Most common texture components in BCC (%)** | | | | |
| Goss | 5.4 | 17.6 | 1.7 | 31.1 |
| Cube | 7.3 | 6.7 | 2.5 | 0.8 |

The annealed and drawn (distorted) specimens were further analysed using STEM on electron-transparent lamellas prepared by FIB, evidencing the strong impact of the processing on the microstructure. The spatial resolution at 200 kV was exploited (typical TEM-STEM accelerating voltages are 60–300 kV, in contrast to the few to tens of kV used in SEM-STEM) to perform a detailed study of the samples. Samples under low-angle annular dark field (LAADF) conditions were characterised, highly sensitive to lattice strain and defects [35,36], thus allowing the identification of crystal domains (i.e., grain boundaries), including nanometric crystal grains. Figure 10 shows LAADF images of 0A and 1C specimens before and after the drawing test, in an area free of residual martensite. The microstructural grain morphology and distribution became apparent in the LAADF images (Figure 10a–d), showing differentiated contrast for the different crystal grains.

According to Figure 10a,c, the annealed samples show micrometre-sized grains (each lamella contained only three or four grains), which evolve towards nanograins due to deep drawing in both cases. Interestingly, these crystal nanograins created after drawing show an elongated morphology, with sizes ranging from tens to hundreds of nanometres, which tend to align parallel to each other (Figure 10b,d).

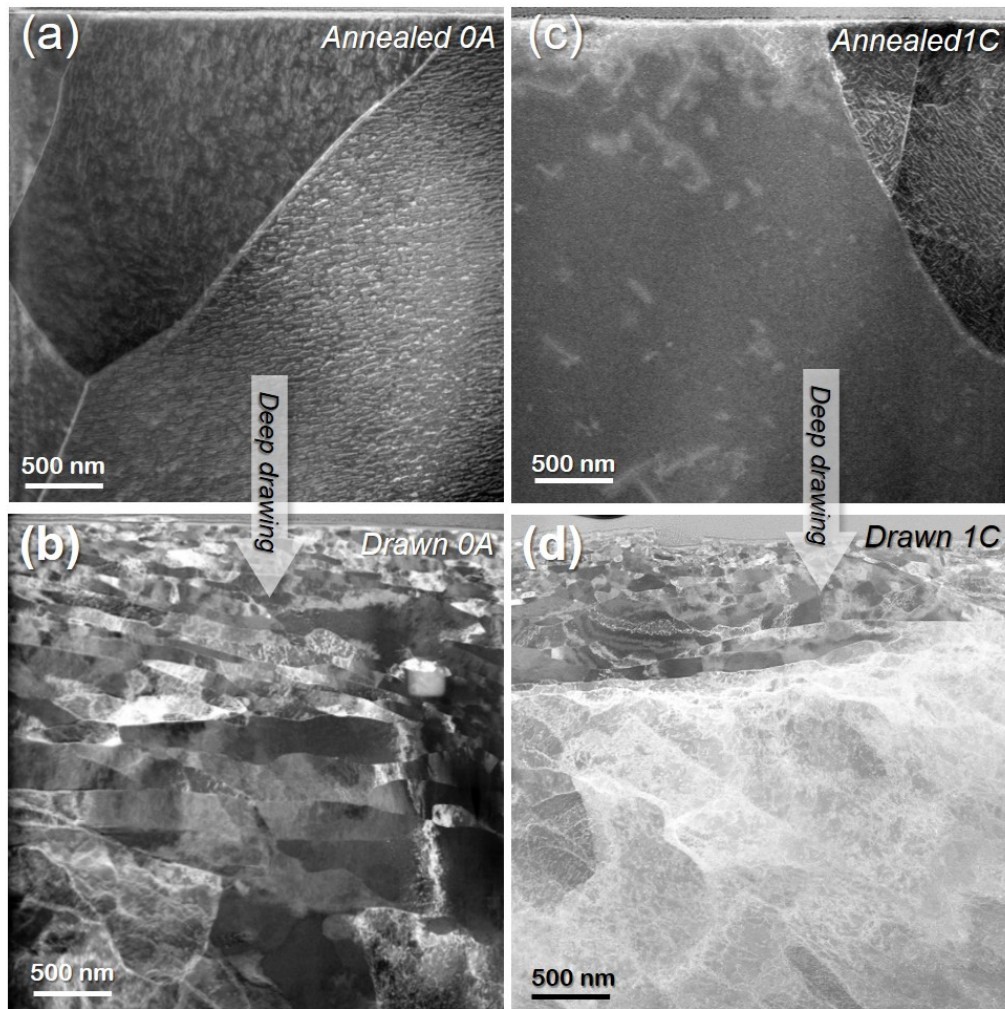

**Figure 10.** LAADF image of 0A specimen (**a**) before and (**b**) after deep drawing. LAADF image of 1C specimen (**c**) before and (**d**) after deep drawing.

A deeper inspection of the LAADF images from the drawn samples provides further insights into the deformation mechanism. Figure 11a,b reveal an ordered grain size distribution within the materials after the drawing tests, meaning that the lateral size of the crystalline grains increases progressively as they move away from the sample's surface, leading to nanograin-sized gradients within each sample. The crystal grains close to the surface are almost equiaxial and reach diameters of tens of nanometres. In contrast, at 500 nm from the surface, they are already larger and highly elongated (heights of several tens of nanometres and lengths of micrometres). Therefore, a likely model of the microstructural distortion induced upon drawing the alloys (Figure 11c) is proposed. Importantly, creating nanograined gradients in metals has recently been reported as an efficient route towards tailoring the alloy properties [37–39], which might play a vital role in the performance of the materials.

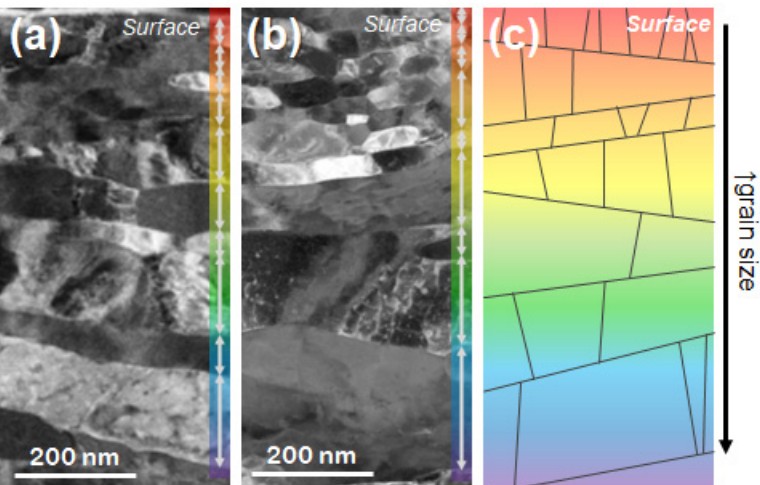

**Figure 11.** LAADF images from deformed 0A (**a**) and 1C (**b**) specimens near the surface. (**c**) Model of the steel microstructural distortion induced upon deep drawing.

## 4. Conclusions

This article presents a comparison between the crystallographic changes that occur during the deep drawing of two types of ferritic stainless steels; the first one is a standard AISI 430 chemical composition (0A), and the second one is a modified ferritic stainless steel (1C) that contains a slight increase in ferrite-stabilising elements and a slight decrease in austenite-stabilising elements. From this, the following conclusions can be made:

- The chemical composition modification in 1C improved the microstructural properties of AISI 430 ferritic alloys. After annealing, 1C displayed an increase in the recrystallisation texture <111>{111} and in the γ-fibre intensity (<111> ||ND, Normal Direction) compared to 0A.
  The annealing performance of 1C is better than 0A for two main reasons. First, during the previous rolling process, 1C underwent a higher level of cold deformation, resulting in a greater concentration of dislocations and higher stored energy prior to annealing compared to 0A. This increase in the stored energy of 1C promotes a larger formation of recrystallisation texture and γ-fibre during the recrystallisation process. This is due to the alloy's higher energy absorption capacity and greater stored energy capacity per unit of mass or volume than 0A.
  The second reason is that the slight changes in the chemical composition of 1C steel have an important impact by increasing its ferritic character at high temperatures. This causes the energy available during annealing to be more likely invested in the recrystallisation process rather than the phase transformation process, as in 0A steel.
- After the deep drawing cup test, the results indicated that the drawability of 1C was better than that of 0A. The crystallographic evolution was different between both alloys. In 0A and 1C, α-fibre (<110>||RD, Rolling Direction) and the Goss component (<001>{110}) appeared. However, in 1C, new deformation components were also identified. The fact that 1C had a stronger α-fibre intensity and higher Goss component indicates that this alloy dissipated more energy during deep drawing than 0A.
- After deep drawing, the most close-packed direction in 0A was <001>{110}. In the case of 1C, this article states that the slip system is defined as <101>{110}, <001>{110}, and <100>{111}. From the structural point of view, deep drawing produces the division of ferrite grains of about 8 to 10 μm (annealing average GS, Grain Size) into nanograins that are aligned parallel to the drawing direction. The deformation mechanism that takes place in ferritic stainless steels during deep drawing can be explained by a distortion model based on the size gradient of the nanograins. Specifically, the model describes the transition of the nanograins from an equiaxed to a columnar structure.

**Author Contributions:** A.N.: Conceptualisation, investigation, and writing—original draft preparation. I.C.: Methodology, resources, and formal analysis. M.D.l.M.: Formal analysis. J.F.A.: Supervision and validation. D.L.S.: Supervision and writing—reviewing and editing. All authors have read and agreed to the published version of the manuscript.

**Funding:** This research received no external funding.

**Data Availability Statement:** The raw/processed data required to reproduce these findings cannot be shared at this time due to legal or ethical reasons.

**Acknowledgments:** The authors acknowledge the Electron Microscopy division at SCCYT-UCA.

**Conflicts of Interest:** I. Collado and J. F. Almagro are employees of the company Acerinox (Spain). The remaining authors declare that the research was conducted in the absence of any commercial or financial relationships that could be construed as a potential conflict of interest.

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
