# Peer review of "A Combined Microscopy Study of the Microstructural Evolution of Ferritic Stainless Steel upon Deep Drawing: The Role of Alloy Composition"

_jmmp, doi:10.3390/jmmp8010006_

Round 1

Reviewer 1 Report

Comments and Suggestions for Authors

The present work entitled "A combined microscopy study of the microstructural evolution of ferritic stainless steel upon deep drawing: the role of alloy composition" considers the study of the texture and microstructure of standard and the modified AISI 430 steel. Deep drawing cup tests 81 were performed to evaluate the drawability and mechanical performance.

The manuscript is generally well written. The methodological part is described in detail. The experiments were well designed. I think this paper can be considered for publication, but it needs some improvements.

1. It is unclear why steel blanks of different thicknesses were used during the Deep Drawing Cup Test. (0.8 and 0.4 mm) This makes the experiment inaccurate.

2. Line 92-94: The proposal is not written entirely clearly.

3. Annealing modes are not specified.

4. Line 199: Ambiguous conclusion about the presence of martensite in the structure.

The paper with opportune modifications can be considered for publication.

Author Response

  1. It is unclear why steel blanks of different thicknesses were used during the Deep Drawing Cup Test. (0.8 and 0.4 mm) This makes the experiment inaccurate.

Response: We agree with the reviewer’s observation, but would like to compare the materials under their most commonly used geometry. Therefore, since 0A steel is usually manufactured from 0.5 to 0.8 mm, whereas 1C steel does from 0.35 to 0.5 mm. These comments are included in the revised text, lines 108-110.

  1. Line 92-94: The proposal is not written entirely clearly.

Response: We thank the reviewer to point this issue. The proposal has been modified as follows to make it clearer (lines 102-104): “These materials are identified as 0A, which has the standard AISI 430 chemical composition, and 1C, that shows some modifications in chemical composition and manufacturing with respect 0A”. In addition, the chemical composition of both alloys is now include in a table for ease of comparison, line 114.

  1. Annealing modes are not specified.

Response: We thank the reviewer the observation. These materials were annealed in the non-oxidising atmosphere (H2/N2) at 850-860 ºC. More details about the production process can be found at www.acerinox.com. These comments are included in the revised text, lines 99-101.

  1. Line 199: Ambiguous conclusion about the presence of martensite in the structure.

Response: We have rewritten the text accounting for the concerns raised by the reviewer, lines 208-242, where also new images about the structure and EBSD maps have been included to clarify this issue: Figure 5, GB+CSL map of the microstructure of the annealed a) 0A and b) 1C specimens obtained by EBSD; Figure 6, a) GB+CSL map of an area with high-twin-concentration of annealed-0A. b) Pattern quality (band slope) map of 0A. c) High-magnification LOM image of an area with high distortion of the crystal lattice, marked with arrows in the image, of annealed-0A. d) Band slope distribution function; and Figure 7, XRD spectra of annealed-0A and 1C, related to the martensite formation. Three new references supporting these results have been added, [31-33] lines 444-449.

Reviewer 2 Report

Comments and Suggestions for Authors

The paper was written in a good shape, and it was well presented and well discussed. It contains useful results and discussion, in particular to the readers who interest in drawing of ferritic stainless steel. However, it has a minor revisions that may be applied to refine the content to avoid the confliction at the readers.

From line 92-97, the authors have listed the chemical composition of ferritic stainless steel that have been studied in this research. However, it is preferable to the reader to show them in a table. 

The authors have mentioned that the samples have been annealed, but they must mention the annealing temperature, which should be useful for the reader.

The authors refer to high twins concentration in Fig.5.b, while the magnification is low for the reader to observe the twins

Author Response

From line 92-97, the authors have listed the chemical composition of ferritic stainless steel that have been studied in this research. However, it is preferable to the reader to show them in a table.

Response: We thank the reviewer to point this issue. The chemical composition listed in this paragraph is showed in a table for ease of comparison, Table 1, line 114.

The authors have mentioned that the samples have been annealed, but they must mention the annealing temperature, which should be useful for the reader.

Response: We thank the reviewer the comment. The typical final annealing temperature for these materials in the non-oxidising atmosphere furnaces (H2/N2) is 850-860 ºC. More details about the production process can be found at www.acerinox.com. These explanations are included in the text, lines 99-101.

The authors refer to high twins concentration in Fig.5.b, while the magnification is low for the reader to observe the twins.

Response: We agree with the reviewer’s observation. A new image with higher magnification is included in Fig. 6 c), line 226, where twins are clearly observed: High-magnification LOM image of an area with high distortion of the crystal lattice of annealed-0A sample.

Reviewer 3 Report

Comments and Suggestions for Authors

I am not entirely sure that the article, entitled (Manuscript ID: jmmp-2762777): „Combined Microscopy Study of the Microstructural Evolution of Ferritic Stainless Steel upon Deep drawing: The Role of Alloy Composition” fits thematically into the Journal of Manufacturing and Materials Processing.

The reviewed article focuses on the microstructure characterisation of two ferritic steels differing in chemical composition and intended for the deep drawing process, and mainly on the EBSD analysis.

The detailed comments and observations are as follows.

Introduction:

The 'Introduction' chapter is poorly written, and contains a lot of general information. Avoid blocks of references, such as this: ‘[...] … Several studies have investigated the texture evolution of FSSs during forming processes using different techniques such as X-ray diffraction (XRD) [8–12], electron backscatter diffraction (EBSD) on scanning electron microscopes (SEM), [8–18] and transmission electron microscopy (TEM) [19]. However, most of these studies focused on the most popular standard grade of FSSs, AISI 430 [11,16], while others explored the texture evolution of modified grades with higher contents of Ti and Nb [8–10,13–15,17–19].. These do not emphasise the particular aspects of the cited papers. In particular, when citations are made about specific technical aspects, single / double references, e.g. [1, 2] are encouraged. It is strongly suggested that the references make in-depth comments on the content of the cited papers.

Results and Discussion

I do not entirely agree with the interpretation of the results. The authors write, for example. […] According to [24] the pattern pole distribution observed in {100} plane of Figure 2a correspond to the formation of martensite in this alloy. …[…] In Figure 5a, the GB+CSL map shows that these fine grains correspond to regions with a high twins concentration (sigma-3). Moreover, in Figure 5b it is verified that there is a high crystallographic distortion in these regions of high-twin-concentration (red-marked). …”

The microstructure images show neither twins nor martensite....The images in Figure 8 confirm that this is not martensite.

It would be useful to analyse the microstructure correctly using X-ray phase analysis.

Some of the Figs., e.g. Figure3, are of very poor quality.

Conclusions should rather avoid abbreviations.

Author Response

Introduction:

The 'Introduction' chapter is poorly written, and contains a lot of general information. Avoid blocks of references, such as this: ‘[...] …

Response: We thank the reviewer the comment, but the references are written according to the Instructions for Authors of the Journal JMMP, as follow: In the text, reference numbers should be placed in square brackets [ ], and placed before the punctuation; for example [1], [1–3] or [1,3]”.

Several studies have investigated the texture evolution of FSSs during forming processes using different techniques such as X-ray diffraction (XRD) [8–12], electron backscatter diffraction (EBSD) on scanning electron microscopes (SEM), [8–18] and transmission electron microscopy (TEM) [19]. However, most of these studies focused on the most popular standard grade of FSSs, AISI 430 [11,16], while others explored the texture evolution of modified grades with higher contents of Ti and Nb [8–10,13–15,17–19]..These do not emphasise the particular aspects of the cited papers. In particular, when citations are made about specific technical aspects, single / double references, e.g. [1, 2] are encouraged. It is strongly suggested that the references make in-depth comments on the content of the cited papers.

Response: The paragraph mentioned by the reviewer summarizes the result of the literature search on texture evolution in FSSs, which includes a total of 12 articles. The authors' intention with this paragraph is to present the reader with an extensive and up-to-date list of articles available in the literature on this particular subject, as well as to highlight the significance of our study. Unfortunately, none of the articles found have a direct correlation with the topic of our study, which is the texture evolution of FSSs before and after deep drawing. Therefore, we deemed it unnecessary to provide further details about them, except for the techniques they have employed and their general approach. However, in response to the reviewer's suggestion, we have chosen a few of the references and provided more detailed comments on them, lines 65-75.

Results and Discussion:

I do not entirely agree with the interpretation of the results. The authors write, for example. […] According to [24] the pattern pole distribution observed in {100} plane of Figure 2a correspond to the formation of martensite in this alloy. …[…] In Figure 5a, the GB+CSL map shows that these fine grains correspond to regions with a high twins concentration (sigma-3). Moreover, in Figure 5b it is verified that there is a high crystallographic distortion in these regions of high-twin-concentration (red-marked). …”

The microstructure images show neither twins nor martensite....The images in Figure 8 confirm that this is not martensite.

It would be useful to analyse the microstructure correctly using X-ray phase analysis.

Response: We have rewritten the text accounting for the concerns raised by the reviewer, lines 208-242, where also new images about the structure and EBSD maps have been included to clarify this issue: Figure 5, GB+CSL map of the microstructure of the annealed a) 0A and b) 1C specimens obtained by EBSD; Figure 6, a) GB+CSL map of an area with high-twin-concentration of annealed-0A. b) Pattern quality (band slope) map of 0A. c) High-magnification LOM image of an area with high distortion of the crystal lattice, marked with arrows in the image, of annealed-0A. d) Band slope distribution function; and Figure 7, XRD spectra of annealed-0A and 1C, related to the martensite formation. Three new references supporting these results have been added, [31-33] lines 444-449.

In Figure 10 (before Figure 8), an area free of residual martensite was analyzed, line 310. These comments are included in the revised text, lines 302-303.

Some of the Figs., e.g. Figure3, are of very poor quality.

Response: We agree with the reviewer’s observation. The quality of Figure 3 has been improved (line 181). The rest of figures have been reviewed.

Conclusions should rather avoid abbreviations.

Response: We thank the reviewer to point this issue. The abbreviations have been detailed as follows: ND, Normal Direction (line 336); RD, Rolling Direction (line 352); GS, Grain Size (line 360).

Round 2

Reviewer 3 Report

Comments and Suggestions for Authors

Dear Authors,

Thank you very much for addressing my remarks and comments. However, there are some minor errors in the article. Detailed comments are presented below:

1. Regarding my comments on the "Introduction" section, my point was more focused on the importance of avoiding a grouping of references. In your case, you repeatedly cite 10 works, for example: “[…] Several studies have investigated the texture evolution of FSSs during forming processes using different techniques such as X-ray diffraction (XRD) [8–12], electron backscatter diffraction (EBSD) on scanning electron microscopes (SEM), [8–18] and….[…] while others explored the texture evolution of modified grades with higher contents of Ti and Nb, which have improved corrosion resistance and weldabilitly [8–10,13–15,17–19].” It is recommended to develop the topic and provide at least a brief description of what the works refer to.

2. On page 4, lines 158-159, there appears to be an issue: “[…] ¡Error! No se encuentra el origen de la referencia.2 shows…” Please correct it.

Best regards,

Author Response

  1. Regarding my comments on the "Introduction" section, my point was more focused on the importance of avoiding a grouping of references. In your case, you repeatedly cite 10 works, for example: “[…] Several studies have investigated the texture evolution of FSSs during forming processes using different techniques such as X-ray diffraction (XRD) [8–12], electron backscatter diffraction (EBSD) on scanning electron microscopes (SEM), [8–18] and….[…] while others explored the texture evolution of modified grades with higher contents of Ti and Nb, which have improved corrosion resistance and weldabilitly [8–10,13–15,17–19].” It is recommended to develop the topic and provide at least a brief description of what the works refer to.

Response: We appreciate the reviewer's point about the style of citing references, which should ideally include a brief description of the works cited. However, as we stated in our previous response, we disagree that the paragraph in question should follow this principle, for two reasons. First, commenting on each of the eleven articles cited in this paragraph would make the introduction too lengthy. Second, these citations are not very relevant to the topic of the article, so summarizing their contents would distract the readers from the main points of the article. We also do not think it would be wise to delete some of these citations, as they form a comprehensive list of references. Lastly, we have thoroughly reviewed the journal's guide for authors and have found no inconsistency between it and our use of references in this paragraph.

  1. On page 4, lines 158-159, there appears to be an issue: “[…] ¡Error! No se encuentra el origen de la referencia.2 shows…” Please correct it.

Response: We thank the reviewer to point this issue. The error is correct it.
